# Continual Learning for Domain Adaptation in Chest X-ray Classification

**Matthias Lenga**                                         Matthias.Lenga@philips.com
**Heinrich Schulz**                                         heinrich.schulz@philips.com
**Axel Saalbach**                                           axel.saalbach@philips.com
*Philips Research Hamburg, Röntgentrasse 24-26, 22335 Hamburg, Germany*

## Abstract

Over the last years, Deep Learning has been successfully applied to a broad range of medical applications. Especially in the context of chest X-ray classification, results have been reported which are on par, or even superior to experienced radiologists. Despite this success in controlled experimental environments, it has been noted that the ability of Deep Learning models to generalize to data from a new domain (with potentially different tasks) is often limited. In order to address this challenge, we investigate techniques from the field of *Continual Learning* (CL) including Joint Training (JT), Elastic Weight Consolidation (EWC) and Learning Without Forgetting (LWF). Using the ChestX-ray14 and the MIMIC-CXR datasets, we demonstrate empirically that these methods provide promising options to improve the performance of Deep Learning models on a target domain and to mitigate effectively *catastrophic forgetting* for the source domain. To this end, the best overall performance was obtained using JT, while for LWF competitive results could be achieved - even without accessing data from the source domain.

**Keywords:** Convolutional Neural Networks, Continual Learning, Catastrophic Forgetting, Chest X-Ray, ChestX-ray14, MIMIC-CXR, Joint Training, Elastic Weight Consolidation, Learning Without Forgetting.

## 1. Introduction

The availability of multiple hospital-scale chest X-ray datasets and the advances in the field of Deep Learning have facilitated the development of techniques for automatic image interpretation. Using Convolutional Neural Networks (CNNs), for multiple findings, performance levels were reported which are on par, or even superior to those of a experienced radiologist. Following the promising results of CNNs for Pneumonia detection in chest X-rays (Rajpurkar et al., 2017), the success of these methods has been transferred to Cardiomegaly, Edema, and Pleural Effusion (Irvin et al., 2019). More recently, for all findings in the ChestX-ray14 dataset (Wang et al., 2017), a performance similar to radiologists was reported (Majkowska et al., 2019). At the same time, it has been noted that these models can be subject to substantial performance degradations when applied to samples from another dataset or domain (Zhang et al., 2019; Yao et al., 2019). In single chest X-ray studies, it is commonly expected that the data is independent and identically distributed among the training and test set, a common assumption in Machine Learning. Contrary, when comparing different publicly available chest X-ray datasets, often a significant domain bias can be observed. Such differences in the data distribution pose a severe challenge for the development, evaluation and validation of medical devices. The occurrence of such

domain-dependent distribution shifts could be, for example, explained by hospital specific processes (including machine protocols and treatment policies), the patient population as well as demographic factors (Zhang et al., 2019). Furthermore, the data collection strategy and the employed labeling techniques impact the specific characteristics of a dataset. The development of domain-invariant predictors has received increased interest, including methods based on bias regularized loss functions and domain augmentation (Zhang et al., 2019) as well as a simultaneous training on multiple datasets (Yao et al., 2019). These approaches have shown great promise to mitigate the effect of a domain shift, but were developed for a one-time optimization prior to model deployment. On the other hand, for most Deep Learning algorithms it is rather straight forward to implement some basic functionality which allows to learn in a continuous fashion (even after deployment) and to improve over time. This includes the adaptation to a new domain and new tasks. This has also be noted by the FDA in a recent discussion about the regulatory implications of modifications to AI/ML-based medical devices, contrary to "locked" software (U.S. Food & Drug Administration, 2019).

Therefore, in this contribution, we approach this challenge from a different perspective i.e. using methods from the field of Continual Learning (CL). Traditionally, in Continual Learning, methods are considered for the sequential learning of individual tasks (Parisi et al., 2019), a concept with great potential for the adaptation of chest X-ray models to a new domain. However, a fundamental problem in CL is catastrophic forgetting (McCloskey and Cohen, 1989), i.e. a phenomenon which is associated with performance degradations for previously learned tasks when a model is adapted to a new task. For chest X-ray classification, this could result not only in a reduced detection performance for unique findings from the source domain, but the model could unlearn to classify data from the source domain in general. Baseline techniques such as Joint Training (JT), try to alleviate this problem by means of integrating data from the source domain into the learning process - an approach which is not always feasible when sensitive healthcare information is considered. Regularization-based CL techniques, such as Elastic Weight Consolidation (EWC) (Kirkpatrick et al., 2017) and Learning Without Forgetting (LWF) (Li and Hoiem, 2017), introduce prior information or soft-targets in order to avoid the need for memorizing old data. In order to evaluate the feasibility of CL techniques, we assess their performance in an empirical study using the ChestX-ray14 and the MIMIC-CXR (Johnson et al., 2019) dataset.

## 2. Material and Methods

In this sections we will provide a brief summary of the three Continual Learning concepts JT, EWC and LWF. The latter two methods follow simple regularization paradigms and do not require the storing of training data from previous tasks or domains. All methods are easy to implement and do not entail a large computational overhead compared to the original model training. Following the conventions from the CL literature, we use a task-centered formalism to describe the CL methods. In our chest X-ray scenario described above, the first and second task correspond to solving the ChestX-ray14 and MIMIC-CXR classification problem, respectively. In the rapidly growing field of CL other methods, for example, relying on episodic memory, generative models or architectural changes of the

network, have been proposed. For a broader overview we refer the interested reader to (Parisi et al., 2019), (Caruana, 1997) and the references therein.

### 2.1. Joint training (JT)

Suppose that $T_i = (x_{i,j}, y_{i,j})_{j=1,...,N_i}$, $i \in I$ is a sequence of tasks where $x_{i,j}$ denotes the $j$-th sample of task $i$ and denotes $y_{i,j}$ the corresponding label. A neural network with weight vector $\theta$ is used to model the predictive distribution $p(y|\theta, x)$ of unobserved labels $y$ associated to observed samples $x$. The model fit is typically conducted by empirical risk minimization. Hence, for each individual task $T_i$, the task-specific optimal weight vector $\theta_i$ is obtained by solving a minimization problem of the type

$$\theta_i = \underset{\theta}{\operatorname{argmin}} \, L(\theta, T_i) := \underset{\theta}{\operatorname{argmin}} \sum_{j=1,...,N_i} - \log p(y_{i,j}|\theta, x_{i,j}). \tag{1}$$

A joint training strategy (JT) aims at improving the model performance on different tasks simultaneously by combining the task-specific training datasets. For example, given a subset of tasks $J \subset I$ the optimal weight vector $\theta_J$ on the combined task $T_J := \cup_{i \in J} T_i$ is obtained by solving a minimization problem of the type

$$\theta_J = \underset{\theta}{\operatorname{argmin}} \, L(\theta, T_J) := \underset{\theta}{\operatorname{argmin}} \sum_{i \in J} \sum_{j=1,...,N_i} - \log p(y_{i,j}|\theta, x_{i,j}) \tag{2}$$

allowing the learning process to exploit commonalities and differences across different tasks. This can improve the predictive performance when compared to training multiple task-specific models separately or training a single model in a simple sequential fashion which is prone to catastrophic forgetting, cf. (Caruana, 1997). Unfortunately, in many real world scenarios, the aggregation of large heterogeneous training datasets (e.g. for chest X-ray classification) is subjected to various limitations. In particular, task-specific data used for model training may no longer be available at some future time point when data associated to a new task is obtained and model fine-tuning becomes necessary.

### 2.2. Elastic Weight Consolidation (EWC)

Various CL approaches for explicitly modeling cross-correlations between distinct tasks have been proposed. Elastic weight consolidation assumes a prior distribution $p(\theta|T_{i-1})$ on the network weights $\theta$ during the model adaptation for task $T_i$. The prior $p(\theta|T_{i-1})$ is selected in such a way that it captures basic statistical properties of the empirical distribution of the network weights across the previous task $T_{i-1}$. Finally, the optimal parameter for the current task $T_i$ is obtained as the maximum a posteriori estimate

$$\theta_i := \underset{\theta}{\operatorname{argmax}} \sum_{j=1,...,N_i} \log p(\theta|x_{i,j}, y_{i,j}) = \underset{\theta}{\operatorname{argmin}} \, L(\theta, T_i) - N_i \log p(\theta|T_{i-1}). \tag{3}$$

In contrast to memory-based methods, EWC acts as a simple regularizer on the training objective and does not rely on storing any additional data associated to previous tasks. The key assumption of EWC is that enough information about previous tasks can be encoded

within the model weight prior distribution in order to prevent a severe performance degradation when moving to a new task. Owing to their computational tractability, frequent choices for $p(\theta|T_{i-1})$ are multivariate Laplace or Gaussian distributions. In the Gaussian case $p(\theta|T_{i-1}) = \mathcal{N}(\theta|\mu_{i-1}, \Sigma_{i-1})$ we obtain

$$\theta_i = \underset{\theta}{\operatorname{argmin}} L(\theta, T_i) + \lambda(\theta - \mu_{i-1})^\top \Sigma_{i-1}^{-1}(\theta - \mu_{i-1}) \tag{4}$$

with a constant $\lambda > 0$ which allows to regulate the impact of the prior. Choosing the parameters $\mu_{i-1} = \theta_{i-1}$ and $\Sigma_{i-1}^{-1} = \operatorname{diag}(F_{i-1})$, where $F_{i-1}$ denotes the empirical Fisher matrix associated to task $T_{i-1}$, i.e.

$$F_{i-1} := \frac{1}{N_{i-1}} \sum_{j=1,\dots,N_{i-1}} \nabla_\theta \log p(y_{i-1,j}|\theta_{i-1}, x_{i-1,j}) \nabla_\theta \log p(y_{i-1,j}|\theta_{i-1}, x_{i-1,j})^\top, \tag{5}$$

yields the EWC objective from (Kirkpatrick et al., 2017). It is well known that under mild regularity assumptions (5) constitutes an approximation to the empirical Hessian of the negative log-likelihood (NLL) with respect to $\theta$, i.e.

$$\mathbb{E}_{y\sim p|\theta,x} H_\theta[-\log p(y|\theta,x)] = \mathbb{E}_{y\sim p|\theta,x} \nabla_\theta \log p(y|\theta,x) \nabla_\theta \log p(y|\theta,x)^\top \tag{6}$$

holds true. Consequently, the entries of $\operatorname{diag}(F_{i-1})$ may be considered as approximations to the non-mixed second derivatives of the NLL, which reflect to some extend the sensitivity of the model output with respect to marginal changes in the network weights. As argued in (Kirkpatrick et al., 2017), second derivatives of large magnitude attribute a high importance of the corresponding model parameter for solving the task $T_{i-1}$. Consequently, the quadratic penalty term in (4) discourages strong deviations from the previous task's parameter $\theta_{i-1}$ in the sensitive weight space directions. In summary, by imposing a prior $p(\theta|T_{i-1})$ on the model weights, deviations from $\theta_{i-1}$ are penalized while learning the task $T_i$ parameter $\theta_i$. The magnitude of the penalty depends of the choice on the prior. For example, prior distributions which are highly concentrated at $\theta_{i-1}$ may severely constrain the flexibility of the model to adapt to the new task $T_i$ in favor of preserving the model performance on $T_{i-1}$. Elastic weight consolidation acts as a regularizer for the current task's model weights and does not require to store the training data from previous tasks.

### 2.3. Learning Without Forgetting (LWF)

The key idea of the Learning Without Forgetting method is to introduce a soft-target regularization into the training loss associated to the current task which reflects the behavior of the model associated to the previous task on the dataset at hand.

In more detail: When moving to a new task $T_i = (x_{i,j}, y_{i,j})_{j=1,\dots,N_i}$ we apply the previous model $M_{\theta_{i-1}}$ which was trained on $T_{i-1}$ to the current task's training samples $x_{i,j}$ in order to generate "synthetic labels" $\hat{y}_{i,j} := M_{\theta_{i-1}}(x_{i,j})$ which record the model behavior. Please note that the raw model outputs $\hat{y}_{i,j}$ correspond, depending on the implementation, to float-valued tensors rather than integer class assignments. By adding a regularization term to the loss functional (1), a bias towards a consistent behavior of the models $M_{\theta_i}$ and $M_{\theta_{i-1}}$

on the current task's training samples is introduced. The task $T_i$ optimal model weight $\theta_i$ is then obtained by solving a minimization problem of the type

$$\theta_i = \operatorname*{argmin}_{\theta} \sum_{j=1,\ldots,N_i} -\log p(y_{i,j}|\theta, x_{i,j}) - \lambda \log p(\hat{y}_{i,j}|\theta, x_{i,j}). \tag{7}$$

Increasing the parameter $\lambda > 0$ decreases the relevance of the "hard-labels" $y_{i,j}$ associated to $T_i$ and instead rewards model output patterns which are consist with the previous model. For a detailed discussion of LWF in the classification setting we refer the reader to (Li and Hoiem, 2017). This basic concept can be implemented and extended in various ways. For example, in the classification setting the soft-target concept can be used to fill missing labels when fine-tuning a model on a new dataset where only partial annotations are available. Similar to EWC, this approach acts as a mere regularizer for the current task's model weights. Access to the previous task's training data is not required.

## 2.4. Datasets

In following we consider the datasets ChestX-ray14 (Wang et al., 2017) and MIMIC-CXR (Johnson et al., 2019). The ChestX-ray14 data was released in 2017 by the NIH Clinical Center and consists of 112120 chest X-ray images (AP/PA) from 30805 patients. The images in the dataset were annotated with respect to 14 different findings using an NLP-based analysis of the radiology reports (with an additional "No Findings" label which is typically not considered).

The MIMIC-CXR dataset (consortium version v2.0.0) consists of X-ray images (DICOM) and radiology reports from the Beth Israel Deaconess Medical Center in Boston. For model training and evaluation, we filtered the DICOM data for AP/PA chest X-ray images (based on the DICOM attributes ImageType, PresentationIntentType, PhotometricInterpretation, BodyPartExamined, ViewPosition and PatientOrientation) resulting in a dataset with 226483 images from 62568 patients. In order to generate annotations, we applied the CheXpert labeler to the impression section of the reports, yielding annotations for 13 findings and a "No Finding" label (Irvin et al., 2019)[1]. In contrast to the ChestX-ray14 dataset, for MIMIC-CXR no official train/test split is available. Therefore, we selected randomly 80% of the patients for training while the remaining 20% were assigned to the test split. For the following experiment, it is assumed that matching labels (including "Effusion" and "Pleural Effusion") represent comparable concepts in both datasets. Consequently, we consider in total 21 labels with 7 unique findings for each dataset and 7 findings occurring in both datasets, see Table 1.

## 2.5. Experimental Design

In order to investigate the impact of a domain shift in the data distribution and the potential benefit of the CL methods outlined in 2.1, 2.2 and 2.3, a set of networks was adapted first to ChestX-ray14 and subsequently to MIMIC-CXR. To this end, a pre-trained DenseNet121 (Huang et al., 2017) was selected as a starting point as it is one of the most commonly employed neural network types in the X-ray domain. In order to account for the changed

---

1. For convenience we adopted the U-Zeroes approach from (Irvin et al., 2019) for uncertain labels.

number of labels and the multi-label classification task, the last layer was replaced by a randomly initialized linear layer and a sigmoid activation function. For the first and second adaptation step a similar hyper-parameter setup was employed: Binary cross entropy was used as a loss function, while for all training scenarios - except LWF - the computation of the loss (training and validation) was restricted to the labels from the current domain. Stochastic gradient descent with momentum was used as update rule, with an initial learning rate of 0.01, a momentum of 0.9 and a mini-batch size of 16. For the adaption to ChestX-ray14 a $L_2$ weight decay of 0.0001 was employed, whereas for the MIMIC-CXR task, weight decay was disabled. After each epoch, the learning rate was reduced by a factor of 10 if the validation loss did not improve. During the training, the images in a mini-batch were subject to data augmentation with a probability of 90%. Our data augmentation included common strategies such as: scaling ($\pm 15\%$), rotation around the image center ($\pm 5°$), translation relative to the image extend ($\pm 10\%$) as well as mirroring along the midsagittal plane (50% chance). Finally, all images were rescaled to $224 \times 224$ pixel in order to match the input size of the DenseNet121 architecture. After training, the network with the lowest validation loss was used for the processing of the test dataset. All experiments were repeated 5 times with resampled validation sets (using 10% of all patients).

The ChestX-ray14 model was adapted on the MIMIC-CXR dataset using four different training strategies:

1. A standard fine-tuning of the networks using the MIMIC-CXR data only.

2. A JT setup where $20\%, \ldots, 100\%$ of the ChestX-ray14 data was included into the adaptation process in addition to the MIMIC-CXR data, cf. Section 2.1.

3. Fine-tuning on the MIMIC-CXR using EWC regularization with a Gaussian prior distribution on the model weights and an impact of $\lambda = 0.001$, cf. Section 2.2.
   For each fold, the mean and the covariance matrix of the prior was calculated based on the associated final model trained on the ChestX-ray14 data. The parameter vector of the ChestX-ray14 model was selected as mean $\mu_{i-1}$ in the EWC objective (4). As inverse covariance matrix we chose the binarized diagonal of the empirical Fisher matrix (5) calculated over all ChestX-ray14 training samples with sensitivity threshold of $\rho = 0.001$. That is to say, we chose $\Sigma_{i-1}^{-1} = \mathrm{diag}(F_{i-1} > \rho)$, where $F_{i-1}$ is defined as in Equation (5). Consequently, all network parameters with a sensitivity below $\rho$ are not affected by the regularization. All other parameters are shrunk towards $\mu_{i-1}$ uniformly with the rate $\lambda$. We found it useful to select $\rho$ based on the distribution of the main diagonal entries of $F_{i-1}$. For example, setting $\rho$ to the 95%-quantile imposes a uniform regularization on the 5% most sensitive network weights. The intuition behind this binarized EWC version is rather simple: we decompose the weight space of the neural network into a subspace containing the sensitive dimensions and its complement. Then a uniform $L_2$-regularization is applied to the weight vector projected on the "sensitive" subspace. Clearly, the computational overhead of this binary EWC is lower compared to classic EWC.

4. Fine-tuning on the MIMIC-CXR data using LWF regularization with an impact parameter $\lambda = 2.0$, cf. Section 2.3. To facilitate the adaption of the model to the new

domain, we applied the LWF penalty only to the 7 labels *not* present in the MIMIC-CXR dataset. Consequently, in the LWF setting all 21 labels from both domains are considered, wheres for EWC and JT-0% only labels from MIMIC-CXR are taken into account. However, the validation loss is always computed on the domain specific validation data containing only 14 labels.

## 3. Results

Our quantitative results in terms of average AUC values for each finding along with their standard deviations are summarized in Table 1. In the upper row the model performance on the ChestX-ray14 dataset is given, while the bottom row corresponds to the performance on the MIMIC-CXR dataset. The left column (Initial) indicates the performance after an initial training on ChestX-ray14, whereas the right columns (JT-0%, JT-20%, ..., LWF) contain the results after the model adaptation to MIMIC-CXR. When applying the models trained on ChestX-ray14 directly to the MIMIC-CXR data, a decreased performance for the classes Cardiomegaly, Edema, Pneumonia and Pneumothorax can be observed. This indicates that the source domain training data is not representative enough for the target domain data distribution. The strongest decrease is observed for Cardiomegaly with a drop from 0.8806 to 0.7603 mean AUC. For the classes Atelectasis, Consolidation and Effusion the performance on the target domain is comparable or even slightly superior, see lower left quadrant of Table 1. As a consequence of the domain shift, the average AUC across all labels decreases from 0.8106 to 0.7833 making model adaptation unavoidable. The lower right quadrant shows that all CL methods achieve a formidable on-domain model performance on the MIMIC-CXR data with average AUC values across all findings ranging from 0.8190 to 0.8257. In particular, this indicates that both regularization approaches (LWF and EWC) still allow for enough flexibility that the model can adjust to the new domain.

However, a simple adaptation to MIMIC-CXR with no CL strategy (JT-0%) leads to a decrease of the mean AUCs on the ChestX-ray14 domain for all classes except Infiltration, Pneumonia and Pneumothorax. The effect of catastrophic forgetting becomes more evident in Figure 1, which depicts the (averaged) Forward (FWT) and Backward-Transfer (BWT) for all findings. These concepts were introduced by (Lopez-Paz and Ranzato, 2017) in order to measure the knowledge transfer across a sequence of tasks.[2] The BWT measures the changes of model performance on a task $T_i$ after adapting to a new task $T_{i+1}$. In detail, for each individual label the BWT is computed by subtracting the task $T_i$ AUC values (prior to adapting the model to $T_{i+1}$) from the task $T_{i+1}$ AUC values. A negative BWT is often associated with catastrophic forgetting. Contrary, a positive BWT is obtained if the performance on the previous task is increased. Similarly, the FWT measures the effect of learning a task $T_i$ on the performance of a future task $T_{i+1}$ which was not seen during training. In detail, for each individual label the FWT is computed by subtracting 0.5 (AUC of random classifier) from the task $T_{i+1}$ AUC values (without adapting the task $T_i$ model to $T_{i+1}$). While the ChestX-ray14 models achieve a moderate FWT on MIMIC-CXR, the low BWT indicates a considerable drop in performance on ChestX-ray14 after the adaptation (JT-0%). Integrating data from ChestX-ray14 into the training on the new domain allows

---

2. In contrast to (Lopez-Paz and Ranzato, 2017) we employed an AUC-based variant of these measures.

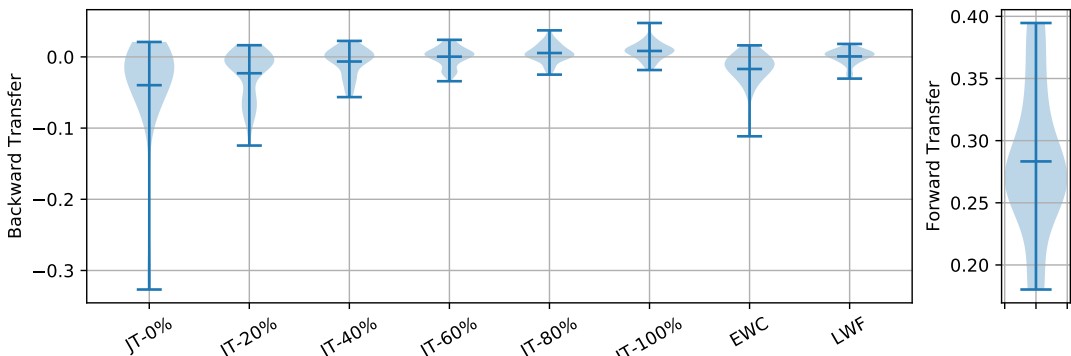

Figure 1: Left: Backward Transfer on ChestX-ray14 after adaptation using different Continual Learning (CL) techniques. Right: Forward-Transfer (FWT) for a chest X-ray14 model on MIMIC-CXR. Bars indicate min, mean and max.

to mitigate this effect (JT-20%,..., JT-100%). We observe that the BWT is positively correlated with the amount of additional samples from ChestX-ray14. Not surprisingly, the best model performance is achieved on the combined dataset containing all training samples from both domains (JT-100%). As argued above, in real world scenarios access to old training data might be limited or not possible at all. Consequently, the regularization based methods LWF and EWC which do not rely on storing data from previous tasks or domains are of high practical relevance. In our experiments, LWF outperformed the EWC approach and achieved a performance on the original domain between JT-60% and JT-80% (and superior to the original model) without accessing any data from ChestX-ray14.

## 4. Conclusion

In this paper we investigated the applicability of different Continual Learning methods for domain adaptation in chest X-ray classification. To that end, a DenseNet121 was trained on ChestX-ray14 and subsequently fine-tuned on MIMIC-CXR using different Continual Learning strategies (JT, EWC, LWF) in order to adapt to the new domain without severe performance degradations on the original data. The motivation for choosing these datasets as distinct domains, was to simulate a realistic domain shift as encountered in clinical practice. Our quantitative evaluation, including the measurement of Backward and Forward Transfer, confirmed that employing these methods indeed improves the overall model performance, compared to a simple continuation of the model training on the new domain. The best performance was achieved by JT-100%, i.e. training the model on the entire combined datasets from both domains. However, in real world scenarios, e.g. adapting models which are already deployed in the clinic, for legal and privacy reasons it is questionable that the data used for training the original model is always accessible. Hence, the EWC and LWF methods which do not rely on old training samples are of high practical relevance. Our experiments indicate that these regularization techniques indeed allow a model adaption to the target domain while preserving a performance on the original domain which is still close to the JT baseline.

|  | Label | Initial | JT-0% | JT-20% | JT-40% | JT-60% | JT-80% | JT-100% | EWC | LWF |
|---|---|---|---|---|---|---|---|---|---|---|
| ChestX-ray14 | Atelectasis* | .7730±.0019 | .7710±.0026 | .7649±.0043 | .7774±.0025 | .7799±.0021 | .7819±.0045 | .7842±.0017 | .7687±.0016 | .7717±.0019 |
| | Cardiomegaly* | .8806±.0021 | .8702±.0058 | .7995±.0101 | .8369±.0090 | .8561±.0058 | .8625±.0048 | .8685±.0046 | .8686±.0046 | .8710±.0052 |
| | Consolidation* | .7468±.0011 | .7393±.0023 | .7461±.0052 | .7526±.0035 | .7541±.0025 | .7537±.0032 | .7572±.0024 | .7353±.0023 | .7386±.0027 |
| | Edema* | .8490±.0010 | .8234±.0028 | .7983±.0036 | .8263±.0068 | .8360±.0070 | .8483±.0022 | .8506±.0053 | .8219±.0038 | .8233±.0027 |
| | Effusion* | .8308±.0009 | .8271±.0013 | .8242±.0024 | .8300±.0016 | .8312±.0012 | .8315±.0015 | .8343±.0014 | .8283±.0021 | .8283±.0024 |
| | Emphysema | .9122±.0054 | .8527±.0096 | .8968±.0022 | .9085±.0029 | .9140±.0050 | .9175±.0030 | .9211±.0033 | .8799±.0061 | .9186±.0059 |
| | Fibrosis | .8273±.0044 | .7458±.0113 | .8145±.0084 | .8286±.0092 | .8325±.0062 | .8350±.0069 | .8370±.0026 | .7869±.0129 | .8309±.0032 |
| | Hernia | .8918±.0174 | .6128±.0743 | .8019±.0351 | .8629±.0067 | .8867±.0127 | .9145±.0079 | .9179±.0055 | .8147±.0356 | .8991±.0168 |
| | Infiltration | .6978±.0011 | .7154±.0029 | .6999±.0056 | .6994±.0039 | .7015±.0025 | .6998±.0035 | .7005±.0040 | .7002±.0156 | .7057±.0036 |
| | Mass | .8203±.0041 | .7826±.0102 | .8160±.0035 | .8256±.0030 | .8281±.0015 | .8328±.0026 | .8355±.0020 | .8078±.0038 | .8294±.0029 |
| | Nodule | .7587±.0035 | .7099±.0104 | .7510±.0031 | .7534±.0048 | .7587±.0030 | .7594±.0020 | .7651±.0041 | .7386±.0063 | .7625±.0036 |
| | Pl. thickening | .7741±.0045 | .7434±.0108 | .7809±.0043 | .7868±.0025 | .7886±.0038 | .7925±.0035 | .7940±.0045 | .7608±.0095 | .7759±.0049 |
| | Pneumonia* | .7233±.0024 | .7261±.0067 | .6549±.0066 | .6879±.0069 | .7033±.0023 | .7112±.0022 | .7142±.0039 | .7318±.0045 | .7288±.0053 |
| | Pneumothorax* | .8620±.0026 | .8721±.0019 | .8774±.0021 | .8807±.0022 | .8822±.0029 | .8834±.0027 | .8843±.0012 | .8668±.0036 | .8738±.0035 |
| | Average | .8106 | .7708 | .7876 | .8041 | .8109 | .8160 | .8189 | .7936 | .8112 |
| MIMIC-CXR | Airspace opacity | | .7735±.0004 | .7750±.0005 | .7741±.0008 | .7744±.0007 | .7745±.0008 | .7748±.0005 | .7708±.0007 | .7741±.0009 |
| | Atelectasis* | .7797±.0019 | .8149±.0007 | .8154±.0006 | .8148±.0010 | .8152±.0011 | .8150±.0005 | .8152±.0011 | .8117±.0011 | .8137±.0006 |
| | Cardiomegaly* | .7603±.0050 | .8305±.0011 | .8301±.0006 | .8297±.0005 | .8302±.0010 | .8300±.0003 | .8299±.0009 | .8284±.0003 | .8310±.0006 |
| | Consolidation* | .7654±.0025 | .8177±.0020 | .8174±.0007 | .8170±.0016 | .8174±.0013 | .8185±.0003 | .8176±.0008 | .8151±.0015 | .8176±.0015 |
| | Edema* | .8343±.0027 | .8904±.0007 | .8909±.0003 | .8900±.0002 | .8905±.0005 | .8899±.0006 | .8902±.0003 | .8890±.0011 | .8904±.0007 |
| | Effusion* | .8913±.0019 | .9200±.0004 | .9205±.0004 | .9202±.0002 | .9204±.0005 | .9204±.0003 | .9204±.0002 | .9190±.0003 | .9202±.0007 |
| | Enl. cardio. | | .6565±.0009 | .6555±.0003 | .6549±.0016 | .6561±.0021 | .6577±.0020 | .6555±.0023 | .6514±.0009 | .6541±.0040 |
| | Fracture | | .7658±.0060 | .7648±.0053 | .7637±.0079 | .7645±.0100 | .7674±.0042 | .7656±.0052 | .7341±.0191 | .7599±.0106 |
| | Lung lesion | | .7998±.0036 | .8055±.0017 | .8018±.0022 | .8057±.0038 | .8063±.0030 | .8077±.0044 | .7959±.0018 | .7999±.0028 |
| | No finding | | .8784±.0007 | .8789±.0004 | .8790±.0006 | .8790±.0005 | .8795±.0005 | .8796±.0003 | .8772±.0006 | .8787±.0007 |
| | Pleural other | | .8629±.0029 | .8633±.0019 | .8633±.0035 | .8659±.0018 | .8645±.0014 | .8657±.0037 | .8585±.0030 | .8612±.0037 |
| | Pneumonia* | .6830±.0031 | .7405±.0014 | .7401±.0021 | .7397±.0011 | .7401±.0006 | .7391±.0023 | .7405±.0019 | .7375±.0019 | .7406±.0019 |
| | Pneumothorax* | .7691±.0064 | .8662±.0023 | .8667±.0017 | .8669±.0023 | .8669±.0023 | .8695±.0015 | .8685±.0020 | .8605±.0032 | .8653±.0018 |
| | Support devices | | .9277±.0015 | .9260±.0013 | .9265±.0024 | .9266±.0015 | .9279±.0009 | .9271±.0020 | .9166±.0027 | .9244±.0033 |
| | Average | .7833 | .8246 | .8250 | .8244 | .8252 | .8257 | .8256 | .8190 | .8237 |

Table 1: Model performance measured in mean AUC and standard deviation on ChestX-ray14 and MIMIC-CXR. Classes marked with a star (⋆) appear in both datasets. **Upper and lower left quadrant**: mean AUCs related to model trained on ChestX-ray14. On-domain performance (ChestX-ray14 model on ChestX-ray14) in upper left, performance on unseen target domain (ChestX-ray14 model on MIMIC-CXR) in lower left. **Upper and lower right quadrant**: mean AUCs related to fine-tuned ChestX-ray14 model on the MIMIC-CXR domain using different Continual Learning strategies - simple fine-tuning without additional data (JT-0%), JT with 20% - 100% inclusion of ChestX-ray14 data, EWC and LWF. On-domain performance (fine-tuned MIMIC-CXR model on MIMIC-CXR) in lower right, performance of original domain (fine-tuned MIMIC-CXR model on ChestX-ray14) in upper right.

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
