# OpenReview forum: "Continual Learning for Domain Adaptation in Chest X-ray Classification"
_MIDL.io/2020/Conference — MIDL 2020_

### Official Review · AnonReviewer4 · 2020-03-11

**Rating:** 2
**Confidence:** 4

**Summary:**

This paper applies some of the current CL methods to a Chest X-ray classification problem with sequential learning on two datasets, involving both domain shift and task shift. The goal is to train a model that performs well on both datasets, including previously seen classes (with potential domain shift) and unseen classes.

**Strengths:**


+ The paper is well-written with clear explanations of previously existing CL methods and experiments.
+ The problem that the authors are trying to solve is practical for potential healthcare applications.

**Weaknesses:**


- The authors used binarized Fisher information for the EWC approach, although a heuristic explanation is given in the last paragraph of page 4, no references or ablation studies were given in the experiments to justify the use of binary EWC regularization.

- Lack of validation on a larger number of tasks/datasets (\eg, 5, 10 tasks), which is typical in CL.

**Detailed Comments:**


1. I think the current title 'Continual Learning for Domain Adaptation' is misleading, which implies the using of CL methods to solve domain adaptation problems. But it's in fact a CL problem with possible involvement of domain shift, \ie, the focus and therefore the evaluation metrics can be very different.

2. Table 1 could be better organized into separated proportions of previously seen classes with domain shift and ChestX-ray14  only classes, which may possibly reveal different degrees of catastrophic forgetting. I would expect the ChestX-ray14  only classes suffer more severely from forgetting.


**Justification Of Rating:**

I vote for borderline. On one side, the authors demonstrate the forgetting problem caused by domain shift, which can be addressed by current regularization-based methods in CL. However, it would be more convincing if the author can validate on a larger number of tasks/datasets instead of using only two datasets to draw a fair conclusion. That is to say, with fixed model capacity, are parameter regularization based CL methods really suitable for practical usage?

**Paper Type:**

validation/application paper

**Special Issue:**

no

---

> ### Author Response · Authors · 2020-03-25
> **Answer to the review**
>
> We would like to thank AnonReviewer4 for the constructive feedback, and the positive assessment of the practical relevance of our work. With regard to the provided comments, we would like to respond as follows:
>
>
> 1) "The authors used binarized Fisher information for the EWC approach [...]"
>
> Answer:
> During the implementation of the EWC algorithm we also investigated a binarized variant.
> While EWC is an established algorithm, we observed that both methods achieved a similar classification performance (with binary EWC being computationally more efficient).
> Therefore, we used binary EWC as a default in our studies. We will elaborate on this observation in the revised version of the manuscript.
>
>
> 2) "Lack of validation on a larger number of tasks/datasets (\eg, 5, 10 tasks), which is typical in CL."
>
> Answer:
> We also noticed that for the presented application techniques from Continual Learning and Domain Adaptation are highly relevant. At the same time, classical CL evaluations (with large number of sequential tasks), might not be suitable as the number of potential pathologies in chest X-ray is rather limited. Therefore, we decided to optimize for multiple task in one update (resembling a deployment of a network in clinical practice).
>
>
> 3) "Table 1 could be better organized [...]"
>
> Answer:
> We completely agree (this point has been also stressed by other reviewers).
> In order to allow for a detailed assessment of the experiments, we decided to provide results for all experiments and pathologies in Table 1 (as well as performance measures such as Forward and Backward Transfer in Figure 1).
> To improve the readability, we will highlight which classes can be attributed to which dataset.
>
>
> 4) "However, it would be more convincing if the author can validate on a larger number of tasks/datasets instead of using only two datasets to draw a fair conclusion"
>
> Answer:
> For the evaluation of the presented methods, we employed two of the largest available chest X-ray datasets (in 5-fold cross-validation). While additional evaluations on new datasets could provide additional evidence, we think that the existing experiments already demonstrated the challenges of catastrophic forgetting and the potential of techniques such as LWF.

---

### Official Review · AnonReviewer1 · 2020-03-14
**An interesting strategy for domain adaptation**

**Rating:** 3
**Confidence:** 5
**Recommendation:** Poster

**Summary:**

Continual learning is known to retain the old knowledge base and use it to learn the new coming tasks. However, in this paper, the author adopt it in a different way that help generalization to data from a new domain. The author used the standard EWC and LwF as the CL methods and test on ChestX-ray14 and MIMIC-CXR datasets.  The author empirically demonstrates that these methods provide promising options to improve the performance on a target domain and to mitigate effectively catastrophic forgetting for the source domain.

**Strengths:**

The author proposed a very interesting idea for domain adaptation, which is to utilize the continual learning methods to use the old knowledge base to help different domain task learning.

The paper is well written and organized.

**Weaknesses:**

1. I would suggest the author can clearly state which algorithm is the proposed and which algorithm is the prior art. For example, put JT, EWC and LwF in the Preliminaries and only describe your method in method.

2. EWC and LwF are a little bit out of age, there are many latest works proposed.
GEM (2017) [1] Lopez-Paz, D., & Ranzato, M. A. (2017). Gradient episodic memory for continual learning. In Advances in Neural Information Processing Systems (pp. 6467-6476).
DEN (2018) [2] Yoon, J., Yang, E., Lee, J., & Hwang, S. J. (2017). Lifelong learning with dynamically expandable networks. arXiv preprint arXiv:1708.01547.
MWC (2019) [3] Zhang, J., & Wang, Y. (2019, October). Continually Modeling Alzheimer’s Disease Progression via Deep Multi-order Preserving Weight Consolidation. In International Conference on Medical Image Computing and Computer-Assisted Intervention (pp. 850-859). Springer, Cham.

And all these methods have released their source code, did the author try their methods on the dataset? Or could the author give more intuition about why to choose EWC and LwF.


3. In EWC, "Fisher matrix calculated over the ChestX-ray14 training samples", I wonder whether the author does the ablation study on how many samples should be used for computing the Fisher matrix will yield the best results.

4. I was confused by the experimental setting "4" :  "The LWF penalty was only applied to the 7 labels
not present in the MIMIC-CXR dataset", can the author make it more clear, is that means the LWF soft-label only compute on the classes in ChestX-ray14? For the remaining 14 classes on MIMIC, there is no LWF loss applied?



**Justification Of Rating:**

This paper proposed a very interesting idea and will become a new strategy for solving domain adaptation on many medical image problem. The paper is well written and organized.  I would give a weak accept and will give my final decision based on the author's feedback in rebuttal.

**Paper Type:**

methodological development

**Questions To Address In The Rebuttal:**

1. I would suggest the author can clearly state which algorithm is the proposed and which algorithm is the prior art. For example, put JT, EWC and LwF in the Preliminaries and only describe your method in method.

2. EWC and LwF are a little bit out of age, there are many latest works proposed.
GEM (2017) [1] Lopez-Paz, D., & Ranzato, M. A. (2017). Gradient episodic memory for continual learning. In Advances in Neural Information Processing Systems (pp. 6467-6476).
DEN (2018) [2] Yoon, J., Yang, E., Lee, J., & Hwang, S. J. (2017). Lifelong learning with dynamically expandable networks. arXiv preprint arXiv:1708.01547.
MWC (2019) [3] Zhang, J., & Wang, Y. (2019, October). Continually Modeling Alzheimer’s Disease Progression via Deep Multi-order Preserving Weight Consolidation. In International Conference on Medical Image Computing and Computer-Assisted Intervention (pp. 850-859). Springer, Cham.

And all these methods have released their source code, did the author try their methods on the dataset? Or could the author give more intuition about why to choose EWC and LwF.

3. In EWC, "Fisher matrix calculated over the ChestX-ray14 training samples", I wonder whether the author does the ablation study on how many samples should be used for computing the Fisher matrix will yield the best results.

4. I was confused by the experimental setting "4" :  "The LWF penalty was only applied to the 7 labels
not present in the MIMIC-CXR dataset", can the author make it more clear, is that means the LWF soft-label only compute on the classes in ChestX-ray14? For the remaining 14 classes on MIMIC, there is no LWF loss applied?


**Special Issue:**

no

---

> ### Author Response · Authors · 2020-03-25
> **Answer to the review "An interesting strategy for domain adaptation"**
>
> We highly appreciate the feedback from the reviewer regarding our study concerning continual learning / domain adaption in chest X-ray classification. With regard to the comments and suggestions of the reviewer we would like to respond as follows:
>
>
> 1) "I would suggest the author can clearly state which algorithm is the proposed and which algorithm is the prior art." AND 2) "EWC and LwF are a little bit out of age, there are many latest works proposed...."
>
> Answer:
> We also see that continual learning is a very active and broad field of research. In this contribution we would like to demonstrate:
>  a) the problem of catastrophic forgetting (which is rarely considered in chest X-ray classification)
>  b) the applicability of “regularization-based” techniques.
>
> For b) we considered EWC and LWF as a starting point (with our binary EWC variant as a minor extension of the classical EWC).
> In this context especially GEM would have been a very interesting extension, but it requires the storage of data from previous tasks (like JT) which we would like to avoid due to privacy concerns.
> We will revise the introduction to emphasize the scope and to motivate the selection of the methods.
>
>
> 3) "In EWC, 'Fisher matrix calculated over the ChestX-ray14 training samples', I wonder whether the author does the ablation study on how many samples should be used for computing the Fisher matrix will yield the best results. "
>
> Answer:
> The Fisher matrix has to be computed just once (after training). Therefore the computational overhead of this step is rather small compared to the overall training.
> At the same time the "statistically best" (~ minimum variance) approximation of the matrix is obtained using the entire dataset (i.e. using many datapoints in the point estimator).
> Therefore, for all experiments the available training data was used for the computation of the matrix.
> In the revised version of the document we will clarify this point.
>
>
> 4) "The LWF penalty was only applied to the 7 labels not present in the MIMIC-CXR
>
> Answer:
> You are completely correct with respect to your comment regarding the LWF soft label.
> In order to avoid catastrophic forgetting, the penalty has been computed only for the classes which are exclusively available in NIH.
> A corresponding clarification will be added to the document.

---

### Official Review · AnonReviewer3 · 2020-03-14
**Nice comparison of methods to mitigate domain shift**

**Rating:** 3
**Confidence:** 4
**Recommendation:** Poster

**Summary:**

The authors present a comparison of methods developed to mitigate the effect of Domain Shift between images acquired with different CXR scanners. All the methods compared have been recently developed and their results are promising, for this reason their application and evaluation in different applications in the field of medical image processing is highly recommended before any further analysis.
The authors perform this work for CXR with extensive and well known datasets in an orderly way.
It is evident that the methodology followed responds to processes more engineering than purely scientific for the value of this type of work for the community more linked to the field of development of new methodologies is undeniable


**Strengths:**


- The comparison methodology in kind of rigurosos
- The methods used for comparison are those most accepted by the community and those that perform best overall. This fact can speed up the work of a large part of the community working on CXR images processing
- The employed datasets and processioning steps are widely accepted by the community
- Application to clinical problems is straightforward



**Weaknesses:**


- I miss a slightly more scientific and less engineering approach
- Annotations,  for MIMIC are obtained employing CheXpert but the possible errors introduced by this have not been taken into account
- Only one architecture is tested (DenseNet121). Even when this architecture is widely accepted for this task would be necessary to extend the study to evaluate the model effect.

- Since the work consists of comparing methods for the DA problem they could have included more datasets (domains). For CXR there are numerous possibilities


**Justification Of Rating:**

The article is well organised and written. It is very useful to compare the different methods of the state of the art and also uses datasets widely used by the community so an analysis like the one made can be very useful for a large number of members of the scientific community in the field of medical image processing.

**Paper Type:**

validation/application paper

**Questions To Address In The Rebuttal:**


- Limit the introduction and list the reasons why this work is useful. There is a small excess of idle talk
- The brief description of the methods is necessary but since the article should be limited to a number of pages and no new methodology is being presented, it would be much clearer to shorten the text and introduce a figure, probably similar to that of LwF paper, that would give the reader a general idea of the similarities and differences between the compared methods at a single glance. Please include this figure in the methods section even if you have to shorten the text.
- Could you provide more details about the image filtering at MIMIC-CXR dataset?
- How could you handle the uncertainty in the annotation introduced by CheXpert?
- Could you show some results with at least one different model architecture?
- Could you include some results, transferring MIMIC to a smaller dataset and from a different problem?
- Could you give more details about the hyper-parametrization process of each method and its training? weight decay and learning rate policies, $\lambda$'s at the regularizers, etc.
- Table 1 is painful to read. Please employ some markers, colours or whatever to ease this task.



**Special Issue:**

no

---

> ### Author Response · Authors · 2020-03-25
> **Answer to the review "Nice comparison of methods to mitigate domain shift"**
>
> We would like to thank the reviewer for the extensive feedback on our paper. We tried to address his/her comments in detail. Please find our anwers below:
>
>
> 1) “Annotations, for MIMIC are obtained employing CheXpert but the possible errors introduced by this have not been taken into account”
>
> Answer:
> In fact, label noise, due to the use of automated report analysis techniques, is a well known problem in all (large scale) chest X-ray classification studies.
> In a comparative evaluation, the CheXpert labler showed a superior performance (e.g. compared to the NIH labler) [IRV],
> while it has been successfully employed for the annotation of the MIMIC-CXR database in previous publications [RBN].
> Based on these findings, and our assessment, we believe that the annotations are sufficient accurate for the evaluation of different
> continual learning techniques
>
>
> 2) “Could you show some results with at least one different model architecture?”
>
> Answer:
> We conducted initial experiments with the PNAS, ResNet and EfficientNet architectures.
> However, the quantitative results differed only slightly, while the message presented in our paper
> remained valid for all settings.
> Since the focus of this paper is to demonstrate that rather simple and generic regularization-based
> CL methods can be applied to successfully tackle the presented continual learning problem on large
> real-world CRX data sets with a strong domain shift, we simply chose the generic and well accepted
> DenseNet architecture.
>
>
> 3) “Could you include some results, transferring MIMIC to a smaller dataset [...]“
> AND “[...] they could have included more datasets“
>
> Answer:
> That is indeed a very valid point. The presented experiments can be extended to include other domains.
> In the literature, the impact on the model performance when switching between MIMIC, ChestX-ray14,
> CheXpert and also other domains has been already studied (without continual learning methods),
> e.g. see the MIDL2019 contribution [YAO].
> A more comprehensive survey including more domains would constitute a natural extension of this paper.
> As stated above, we aimed at providing a generic starting point for such endeavors.
>
>
> 4) “Limit the introduction and list the reasons why this work is useful. There is a small excess of idle talk”
>
> Answer:
> Thank you for your feedback, we will revise the introduction in final version of the paper.
>
>
>
> 5) “[...] Please include this figure in the methods section even if you have to shorten the text.”
>
> Answer:
> A good, unified visualization of EWC and LWF would be nice.
> We already had a look at the visualizations provided in the original LWF and EWC papers.
> Unfortunately, depicting EWC by a pictogram as in the LwF paper (Fig 2) seems to
> us not very informative. Moreover, depicting LWF in a similar manner as in the EWC paper (Fig. 1)
> seems to be cumbersome as well.
>
> Could you please provide us with some more details which kind of visualization
> you would prefer for better illustration?
>
>
> 6) “Could you provide more details about the image filtering at MIMIC-CXR dataset?”
>
> Answer:
> Since the MIMIC-CXR data is not restricted to (AP/PA) chest X-ray images as the ChestX-ray14 dataset,
> the images were filtered based on the DICOM information. While it cannot be assumed that the DICOM
> information is reliable for all images, the filtering step allowed for the exclusion of a substantial amount of images with non-diagnostic information, images containing other body parts or images with a different view position. Our inclusion criteria are:
>
> ImageType: [ORIGINAL, PRIMARY], [DERIVED, PROMARY]
> PresentationIntentType: FOR  PRESENTATION
> PhotometricInterpretation: MONOCHROME1, MONOCHROME2
> BodyPartsExamined: PORT CHEST, CHEST
> ViewPosition: AP,PA
> Pat.Orientation: LF, RF
>
> We will rewrite the section in order to make this point clear.
>
>
>
> 7) “How could you handle the uncertainty in the annotation introduced by CheXpert?”
>
> Answer:
> We adopted the U-Zeroes approach from (Irvin et al., 2019) for the uncertain label.
> Please see the footnote on page 5 of our paper.
>
>
> 8) “Could you give more details about the hyper-parametrization process of each method and its training?"
>
> Answer:
> The hyper-parameters (optimizer, LR-policy, augmentation, etc ..) are described in Section 2.5 of the paper. For each algorithm the same settings have been employed. We will re-write the section to emphasize this aspect.
>
>
> 9) “Table 1 is painful to read. Please employ some markers, colours or whatever to ease this task.”
>
> Answer:
> Thank you for sharing this observation (which has also been noted by other reviewers).
> We will add markers in order to indicate the dataset specific classes.
>
>
> References:
> [YAO] Yao et al.: A Strong Baseline for Domain Adaptation and Generalization in Medical Imaging
> [IRV] Irvin et al.: CheXpert: A Large Chest Radiograph Datasetwith Uncertainty Labels and Expert Comparison
> [RBN] Rubin et al.: Large Scale Automated Reading of Frontal and Lateral Chest X-Rays using Dual Convolutional Neural Networks

---

### Official Review · AnonReviewer2 · 2020-03-15

**Rating:** 3
**Confidence:** 4

**Summary:**

In this paper, the authors investigate the applicability of different Continual Learning methods for domain adaptation in chest X-ray classification. They include joint training (JT), Elastic Weight Consolidation (EWC) and Learning without Forgetting (LWF) in the continual learning analysis on two chest X-ray datasets, ChestX-ray14, MIMIC-CXR.

**Strengths:**

1) The paper clearly describes several continual learning methods, and how they design the experiments based on different methods.
2) The paper is well organized and the design of experiments is reasonable. The results of the experiments are well explained and analyzed.

**Weaknesses:**

1) I am curious about what the results will be if the order of the two tasks are switched that training with MIMIC-CXR first and then adapting to ChestX-ray14. Since MIMIC-CXR has more images than ChestX-ray14, I am wondering if the number of the training samples could have any effect on continual learning.
2) It would be nice if there is some comparison with state-of-the-art methods.

**Justification Of Rating:**

The idea and motivation of the paper is interesting. The paper is well organized and presented. The experiments are reasonable and the results are analyzed well. Thus, I suggest an accept or a weak accept for this paper.

**Paper Type:**

validation/application paper

**Special Issue:**

no

---

> ### Author Response · Authors · 2020-03-25
> **Answer to the review**
>
> First of all, we would like to thank the reviewer for his/her constructive feedback and the provided suggestions for improvement. Please, find our answers below:
>
>
>
> 1) “I am curious about what the results will be if the order of the two tasks are switched that training with MIMIC-CXR first and then adapting to ChestX-ray14. Since MIMIC-CXR has more images than ChestX-ray14, I am wondering if the number of the training samples could have any effect on continual learning.”
>
> Answer:
>
> This is indeed an interesting question! We conducted such an experiment _without_ cross validation Qualitatively,
> the “source domain MIMIC, target domain ChestX-ray14”-setup gives similar results, i.e. that the CL methods
> LWF and EWC perform quite nicely compared to the Joint Training baseline without relying on storing data
> (gradients, etc.) from the target domain. We think that presenting quantitative results for the “switched task”
> scenario would be interesting but comes at the cost of obscuring the exposition without providing any
> further new substantial insights. Instead we decided to provide a detailed evaluation for multiple algorithms
> including a cross-validation study.
>
> In the literature, the impact on the model performance when going from MIMIC to ChestX-ray14
> (and also to other domains) has been studied (without continual learning methods).
> In this regard we would like to refer the interested reader, for example, to the MIDL2019 contribution [YAO].
>
>
>
> 2) “It would be nice if there is some comparison with state-of-the-art methods.”
>
> Answer:
>
> We focused on EWC and LWF on purpose due to various reasons:
> a)	They can be easily integrated into any training setup and do introduce complicated hyper parameters
> b)	The computational overhead during adaptation to the second domain is rather low
> c)	They straightforwardly generalize to image segmentation tasks (would be interesting to try out!)
> d)	They do not rely on storing data (e.g. images or gradient information) related to individual images from the target domain.
>
> In view of [FDA] point d) may be of particular importance.
>
> Many state-of-the art CL methods do not satisfy all criteria a) b) c) d) at once.
> For example: [GEM] requires storing data from the source domain and solving a quadratic problem in each gradient step.
>
> Nonetheless, an extensive study of memoryless methods for continual learning problems on large real-world
> clinical datasets would be interesting for future work. This paper is just a first step.
> Certainly, we did not aim at providing the best neural network architecture along with the most
> suited CL method to solve the specific continual learning problem which arises when moving from
> ChestX-ray14 to MIMIC-CXR.
> Here, the take home message would rather be: choosing a rather generic architecture (such as a DenseNet)
> and a generic CL method such as LWF satisfying our requirements a) – d) already works pretty good!
>
>
> References:
> -----------------
> [FDA] U.S. Food & Drug Administration. Proposed Regulatory Framework for Modiﬁcations to Artiﬁcial Intelligence/Machine Learning (AI/ML)-Based Software as a Medical Device (SaMD). 2019.
> [GEM] David Lopez-Paz, Marc'Aurelio Ranzato: Gradient Episodic Memory for Continual Learning
> [YAO] Yao et al.: A Strong Baseline for Domain Adaptation andGeneralization in Medical Imaging

---

### Meta-Review · Area_Chair1 · 2020-03-31
**MetaReview of Paper37 by AreaChair1**

**Rating:** 3
**Recommendation For Accepted Papers:** Poster

**Metareview:**

The paper has obtained mixed reviews：3 weak accept (R1-R3) and 1 weak reject (R4). All reviewers appreciate the novel idea of applying continual learning to address a domain adaptation problem. The most negative R4 points out 'Lack of validation on a larger number of tasks/datasets', which the AC thinks that author makes a good rebuttal. Therefore, the ACs decide to downplay R4's comments and recommend to weak-accept the paper.

**Paper Type:**

validation/application paper

**Special Issue:**

no

---

### Decision · Program_Chairs · 2020-04-11

Accept